# Factors affecting quality of life in connective tissue disease-related interstitial lung disease

Lanier L. O'Hare[1]*, Anand S. Iyer[1,2], Kathleen O. Lindell[3], Pariya L. Wheeler[4], Liang Shan[4], Tracy Luckhardt[1], Marie Bakitas[2,4]

1 Division of Pulmonary, Allergy, and Critical Care Medicine, Department of Medicine, University of Alabama at Birmingham, Birmingham, Alabama, United States of America, 2 Center for Palliative and Supportive Care, Division of Gerontology, Geriatrics, and Palliative Care, Department of Medicine, University of Alabama at Birmingham, Birmingham, Alabama, United States of America, 3 School of Nursing, Medical University of South Carolina, Charleston, South Carolina, United States of America, 4 School of Nursing, University of Alabama at Birmingham, Birmingham, Alabama, United States of America

* lohare@uabmc.edu

## Abstract

### Background

Connective tissue disease-related interstitial lung disease (CTD-ILD) results in an unrelenting symptom burden and may progress to death. The morbidity and mortality associated with CTD-ILD likely has a profound impact on individuals' health-related quality of life (HRQOL). The factors associated with HRQOL in other chronic lung diseases have been described, but because of the different clinical and demographic characteristics of CTD-ILD, it is unknown if these same factors are associated with HRQOL in CTD-ILD.

### Research questions

What is the association between patient demographic and disease characteristics, symptoms, and HRQOL in CTD-ILD?

### Study design and methods

A cross-sectional design was used to describe HRQOL in CTD-ILD utilizing a secondary data analysis from the Pulmonary Fibrosis Foundation Patient Registry (PFFPR). Data extracted included demographic (age, gender, and race) and disease characteristics [type of CTD-ILD, duration of disease, forced vital capacity (FVC), supplemental oxygen, immunosuppressant medication use, and pulmonary rehabilitation]. Questionnaires were used to evaluate HRQOL and symptoms of shortness of breath, cough, and fatigue.

**Data availability statement:** All relevant data are within the paper.

**Funding:** The author(s) received no specific funding for this work.

**Competing interests:** The authors have declared that no competing interests exist.

## Results

The majority of participants were female (66%), white (78%), had a disease duration of 1–3 years (30%), had scleroderma (25%). The average age was 61 years and FVC of 67% predicted. The majority of participants were not on supplemental oxygen (62%), taking immunosuppressive medications (66%), or active in pulmonary rehabilitation (89%). Female gender, lower FVC, supplemental oxygen use, pulmonary rehabilitation participation, shortness of breath, cough, and fatigue were all correlated with poorer HRQOL. Shortness of breath mediated the relationships between HRQOL and the factors of gender, FVC, supplemental oxygen use, and pulmonary rehabilitation. Fatigue mediated the relationship between HRQOL and pulmonary rehabilitation.

### Interpretation

Disease severity, symptom burden, gender, and disease treatments are associated with poor HRQOL. Recognition of these factors, treating symptoms, and consideration of palliative care may impact HRQOL in CTD-ILD.

### Introduction

Interstitial lung disease (ILD) is characterized by lung inflammation and fibrosis that results in significant symptomatic and psychosocial burden for individuals with the disease [1]. The global mortality rate for ILD has increased over 50% in the past 10 years and is the 40th most common cause of death [2]. In 2015, there were 18,722 deaths attributed to ILD, 4,856 in Latin America, 1806 in Central Europe, and 1705 in Central Asia [2]. Individuals with ILD suffer a high symptom burden that may have a profound and debilitating effect on health-related quality of life (HRQOL) [3]. While there are many causes of ILD, connective tissue disease (CTD) is responsible for approximately one third of all cases [4]. ILD is considered one of the most serious manifestations of CTD in terms of morbidity and mortality [5]. CTD includes a heterogeneous group of rheumatological conditions [6]. CTD is accompanied by rheumatological-related symptoms; hence, individuals with connective tissue disease-related interstitial lung disease (CTD-ILD) may also experience more symptoms that may affect HRQOL more than those with ILD alone [7].

Other lung diseases, such as chronic obstructive pulmonary disease (COPD) and lung cancer have been shown to have a negative impact on HRQOL; however, these diseases do not have the extra burden of rheumatoid-related symptoms [8,9]. Therefore, the impact of CTD-ILD on HRQOL may be more profound, but this has yet to be sufficiently described. The paucity of information about HRQOL in CTD-ILD may be attributed to the heterogeneity of phenotypes associated with the disease [10]. Because of the heterogeneity associated with CTD-ILD, there may be additional factors affecting HRQOL in this population. There are several unique demographic and disease characteristics of CTD-ILD that differentiate it from other chronic lung diseases. CTD-ILD is more prominent in women, with a ratio of 75:25, while in IPF,

the ratio is 25:75 [11]. The female preponderance in CTD-ILD differs from COPD and lung cancer, in which more men have the disease [12,13].

The purpose of this study was to investigate the HRQOL and associated factors as well as the mediating effect of symptoms for individuals with CTD-ILD by performing a secondary data analysis from the Pulmonary Fibrosis Foundation Patient Registry (PFFPR).

## Methods

### Study population and design

Using a secondary data analysis at a single time point from the PFFPR, this study used a cross-sectional design to describe HRQOL in CTD-ILD. The parent study, entitled "Pulmonary Fibrosis Foundation Patient Registry," utilized convenience sampling of individuals with ILD recruited during a clinic visit at one of the 42 participating Pulmonary Fibrosis Foundation Care Centers from March 29, 2016, through data access on April 6, 2022. The authors did not have access to information that could identify individual participants during or after data collection. The inclusion criteria required consented participants to be at least 18 years old, have a confirmed ILD diagnosis, and be available for follow up at the Registry Center for at least one year. Exclusion criteria were inability to complete the study instruments and diagnosis with sarcoidosis, lymphangioleiomyomatosis, pulmonary alveolar proteinosis, cystic fibrosis, or amyloidosis. The PFFPR captured data elements at baseline and then every 6 months during the study [14]. The data elements included demographics, diagnostic information, pulmonary function test results, pulmonary rehabilitation utilization, medication usage, medical event and mortality data, and patient reported outcome measures (PROMs). The immunosuppressive agents included in this study are abatacept, adalimumab, azathioprine, belimumab, cyclophosphamide, cyclosporin A, etanercept, golimumab, hydroxychloroquine, infliximab, leflunomide, methotrexate, mycophenolate, rituximab, sulfasalazine, tacrolimus, and tocilizumab. PROMs are described in Table 1 and include HRQOL [Medical Outcome Study Short Form Six-Dimension Health Survey (SF-6D)], SOB [California San Diego Shortness of Breath Questionnaire (SOBQ)], cough [Leicester Cough Questionnaire (LCQ)], and fatigue [Fatigue Severity Scale (FSS)].

The current study included a subset of ILD cases from the PFFPR database diagnosed with CTD-ILD. CTD-ILD cases were identified as subcategories by name of disease. Alternate forms of ILD were excluded for this study. Excluded cases were those in which questionnaires, demographic information, and FVC data were not completed at baseline. As of April 5, 2022, 333 CTD-ILD cases had complete data for this study. This sample size of 333 meets the requisite 119 cases for twelve predictor variables with a medium significance level of .05 [19].

**Table 1. Variable Measures.**

| Variable | Instrument | Items; Domains; Score |
|---|---|---|
| Quality of life | Medical Outcomes Study Short-Form Six-Dimension (SF-6D) | 11 items; 6 domains: physical functioning, role participation (physical and emotional), social functioning, bodily pain, mental health, and vitality; summary score: 0.3 to 1.0 with 0.3 indicating worst possible quality of life [15] |
| Shortness of breath | University of California San Diego Shortness of Breath Questionnaire (SOBQ) | 24 items; 2 domains: shortness of breath with activities of daily living and limitations due to shortness of breath; 6-point scale, scores ranging from 0–120, with a higher score indicating greater severity of shortness of breath [16] |
| Cough | Leicester Cough Questionnaire (LCQ) | 19 items; 3 domains: social, psychological, and physical; 3–21 score range with a higher score indicating a better HRQOL [17] |
| Fatigue | Fatigue Severity Scale (FSS) | 9 items; 7-point Likert scale, scores are averaged for a composite score, then divided by 9: minimum score is 1.0, maximum score is 7.0 with higher score indicative of higher severity of fatigue [18] |

Investigation of HRQOL in CTD-ILD was guided by the research questions and hypotheses in <u>Table 2</u>, as informed by the conceptual model depicting the factors that affect HRQOL (<u>Fig 1</u>).

Conceptual model of the direct effect of the antecedent factors (demographic and disease characteristics), and symptoms (shortness of breath, cough, and fatigue) on the consequence of quality of life. Mediation effect of symptoms between demographic/disease characteristics and quality of life.

The Ethics Committee of the University of Alabama at Birmingham waived the need for ethics approval and patient consent for the collection, analysis and publication of the retrospectively obtained and anonymized data for this

**Table 2. Research Questions and Corresponding Hypotheses.**

| | |
|---|---|
| Q1 | What is the association between patient demographic (age, gender, and race) and disease characteristics (type of CTD-ILD, duration of disease, FVC, supplemental oxygen, immunosuppressant medication use, and pulmonary reha-bilitation) and HRQOL in CTD-ILD? |
| | H 1.1 Demographic characteristics (younger age, female gender, and Black race) are associated with worse HRQOL. |
| | H 1.2 Disease characteristics (CTD-ILD type, longer duration of disease, lower FVC, immunosuppressant medication use, supplemental oxygen use, and not participating in pulmonary rehabilitation) are associated with worse HRQOL. |
| Q 2 | What is the relationship between symptoms (shortness of breath, cough, and fatigue) and HRQOL in CTD-ILD? |
| | H 2.1 Increased shortness of breath is associated with worse HRQOL. |
| | H 2.2 Increased cough is associated with worse HRQOL. |
| | H 2.3 Increased fatigue is associated with worse HRQOL. |
| Q 3 | What are the potential mediation relationships between symptoms and HRQOL in CTD-ILD? |

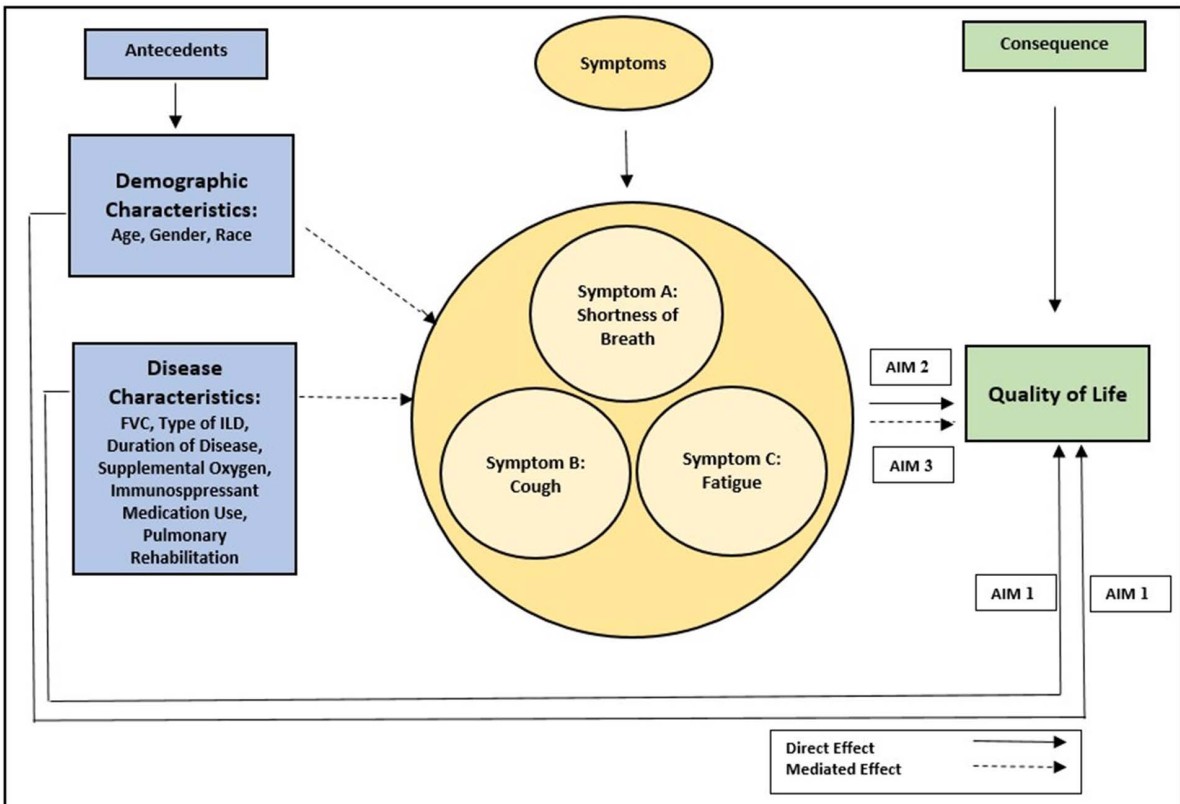

**Fig 1. Factors Affecting Quality of Life Model.**

non-interventional study on March 17, 2022. With approval by both the PFFPR Scientific Review Committee and the University of Alabama at Birmingham Institutional Review Board (300008870), the de-identified data were transferred electronically via encryption.

## Statistical approach

Means and standard deviations were obtained for the continuous variables, including age (years), FVC (in liters), SF-6D, SOBQ, LCS, and FSS. Frequencies and percentages of the categorical variables (race, gender, type of CTD-ILD, disease duration, supplemental oxygen, immunosuppressant medication use, and pulmonary rehabilitation) were also obtained.

Bivariate analysis between the antecedents (demographic characteristics and clinical characteristics) and the potential mediators (symptoms) and the dependent variable of HRQOL was performed using different techniques. The strength of the linear relationships between the continuous variables, including antecedents and symptoms, and HRQOL was examined using Pearson correlation coefficient. Analysis of variance (ANOVA) or two-sample t test was performed to evaluate the strength of relationship between the categorical variables (antecedents and symptoms) and HRQOL.

Path analysis was used to test the hypothesized causal pathways between demographic characteristics and HRQOL, disease characteristics and HRQOL, and symptoms with HRQOL, as depicted in Fig 1. The mediating effect of symptoms on the relationship between demographic and disease characteristics on HRQOL was also investigated. A significance level of 0.05 was established.

## Results

### Descriptive analyses

The average age of the sample was 61.2 years old. Most participants were female (220 [66%]) and White (251 [77.5%]), followed by Black (58 [17.9%]). The mean FVC was 67.1 liters (18.6). Most participants had scleroderma (84 [25.2%]), followed by rheumatoid arthritis (77 [23.1%]), inflammatory myopathy (65 [19.5%]), mixed and undifferentiated CTD (59 [17.7%]), Sjogren's syndrome (24 [7.2%]), lupus (16 [4.8%]), and vasculitis (8 [2.4%]). The majority of participants had CTD-ILD for 1–3 years (99 [30%]), followed by 3–7 years (90 [27.3%]), less than 1 year (85 [25.8%]), and >7 years (57 [17%]). Most participants did not require supplemental oxygen (206 [61.9%]), were not taking immunosuppressant medications (220 [66.1%]), and were not enrolled in pulmonary rehabilitation (295 [88.6%]). The mean PROM scores were: SOBQ 44.6 (26.63), LCQ 15.92 (4.43), FSS 4.46 (1.79), and the SF-6D 0.66 (0.11). The descriptive statistics are summarized in Table 3 and bivariate analysis results in Tables 4 and 5.

### Associations between demographics and disease characteristics and HRQOL

Pearson correlation analysis showed that HRQOL was not linearly correlated with age ($r = .09$, $p = .1174$), but did have a positive linear correlation with FVC ($r = 0.13$, $p = .0223$), such that a higher FVC was correlated with better HRQOL with a small effect size (0.13). HRQOL was statistically different based on gender ($t_{304} = -2.899$, $p = .0042$), and supplemental oxygen use ($t_{261} = 3.779$, $p = .0002$), but was not associated with immunosuppressive medication use. Women reported lower HRQOL than men ($d = -0.35$, $p = .0042$). Supplemental oxygen use was associated with poorer HRQOL ($d = 0.43$, $p = .0002$). The results suggest that there was no statistically significant difference in HRQOL due to race or type of CTD-ILD ($\eta^2 = .0046$, $p = .309$; $\eta^2 = .0022$, $p = .3492$, respectively).

Analysis of variance was conducted to assess the association between disease duration and HRQOL. The results suggest that there was no statistically significant difference in HRQOL among disease duration categories ($F_{3,298} = 1.85$, $p = .138$). The two-sample t test was performed to evaluate differences in HRQOL based on pulmonary rehabilitation participation. The results suggest that there was a statistically significant difference in HRQOL with those participating in pulmonary rehabilitation reporting poorer HRQOL, indicating a large effect size ($p < .0001$, $d = 0.48$).

**Table 3. Participant Demographics and Disease Characteristics.**

| Characteristics | |
|---|---|
| Age, years | 61.24 ± 12.46 |
| Gender | |
| Male | 113 (34) |
| Female | 220 (66) |
| Race | |
| White | 251 (78) |
| Black | 58 (18) |
| Asian | 11 (3) |
| Other | 4 (1) |
| Type of CTD | |
| Scleroderma | 84 (25) |
| Rheumatoid Arthritis | 77 (23) |
| Inflammatory Myopathy | 65 (20) |
| Mixed and Undifferentiated CTD | 59 (18) |
| Sjogren's Syndrome | 24 (7) |
| Lupus | 16 (5) |
| Vasculitis | 8 (2) |
| Duration of Disease | |
| <1 Year | 85 (26) |
| 1–3 Years | 99 (30) |
| >3–7 Years | 90 (27) |
| >7 Years | 57 (17) |
| FVC, liters | 67.14 ± 18.60 |
| Supplemental Oxygen | |
| Yes | 127 (38) |
| No | 206 (62) |
| Immunosuppressant Medication Use | |
| Yes | 113 (34) |
| No | 220 (66) |
| Pulmonary Rehabilitation | |
| Yes | 38 (11) |
| No | 295 (89) |
| HRQOL: SF-6D Score | 0.66 ± 0.11 |
| Shortness of Breath: SOBQ Score | 44.60 ± 26.63 |
| Cough: LCQ Total | 15.92 ± 4.43 |
| Fatigue: FSS Total | 4.46 ± 1.79 |

## Relationships of symptoms to HRQOL

We then examined the relationship between symptoms and HRQOL in CTD-ILD, results summarized in Table 5. We found that HRQOL had a positive linear correlation with cough, with higher scores on cough scale indicative of less cough and higher degree of HRQOL ($r = 0.45$, $p < .0001$). HRQOL had a negative correlation with fatigue and SOB such that a lower score on the fatigue scale (indicative of less fatigue) correlated with higher degree of HRQOL ($r = -0.68$, $p < .0001$) and lower scores (indicative of less SOB) correlated with a higher degree of HRQOL ($r = -0.67$, $p < .0001$).

**Table 4. Comparisons of HRQOL by the Categorical Variables (n = 333).**

| Variable | SF-6D HRQOL Score | | |
|---|---|---|---|
| | Mean*(SD) | Effect Size (Cohen's *d* or *η2*) | *p*-value |
| Gender | | $d = -0.35$ | .0042 |
| Male | 0.68 (0.11) | | |
| Female | 0.64 (0.11) | | |
| Race | | $\eta^2 = .0046$ | .3090 |
| White | 0.66 (0.11) | | |
| Black | 0.64 (0.11) | | |
| Asian | 0.61 (0.16) | | |
| Other | 0.60 (0.04) | | |
| Type of CTD | | $\eta^2 = .0022$ | .3492 |
| Scleroderma | 0.64 (0.10) | | |
| Rheumatoid Arthritis | 0.66 (0.11) | | |
| Inflammatory Myopathy | 0.67 (0.12) | | |
| Mixed and Undifferentiated CTD | 0.68 (0.12) | | |
| Sjogren's Syndrome | 0.63 (0.15) | | |
| Lupus | 0.63 (0.12) | | |
| Vasculitis | 0.68 (0.06) | | |
| Duration of Disease | | $\eta^2 = 0.01$ | .1380 |
| <1 year | 0.65 (0.11) | | |
| 1–3 years | 0.65 (0.11) | | |
| >3–7 years | 0.68 (0.12) | | |
| >7 years | 0.64 (0.10) | | |
| Supplemental Oxygen | | $d = 0.43$ | .0002 |
| Yes | 0.63 (0.10) | | |
| No | 0.67 (0.12) | | |
| Immunosuppressant Medication Use | | $d = 0.14$ | .2553 |
| Yes | 0.67 (0.11) | | |
| No | 0.65 (0.11) | | |
| Pulmonary Rehabilitation | | $d = 0.48$ | <.0001 |
| Yes | 0.61 (0.11) | | |
| No | 0.66 (0.11) | | |

*SF-6D scores range from 0.3–1.0, low score indicative or worse HRQOL.

**Table 5. Correlation Coefficients in the Analytical Sample (n = 333).**

| Variable | Pearson Correlation Coefficient with HRQOL |
|---|---|
| 1. HRQOL: SF-6D Score | — |
| 2. Age, Years | 0.09 |
| 3. FVC, Liters | 0.13* |
| 4. Shortness of Breath: SOBQ Score | −0.67** |
| 5. Cough: LCQ Total | 0.45* |
| 6. Fatigue: FSS Total | −0.68** |

*$p < .05$. **$p < .01$.

## Mediating factors affecting HRQOL

Path analysis was used to evaluate potential mediation effect of symptoms on the pathway from the antecedents to HRQOL in CTD-ILD as depicted in the Factors Affecting HRQOL Model (Fig 1). No goodness of fit was evaluated in this study, as the path model is saturated and there is zero degrees of freedom. The results of the mediation analysis are summarized in Table 6. Mediation analysis for symptoms was performed examining the indirect effects of SOB, cough, and fatigue on each antecedent with HRQOL. Gender and FVC had small indirect paths to HRQOL through SOB (standardized coefficient of indirect effect = 0.057, 0.08; p-values 0.011, 0.003, respectively). Supplemental oxygen use and pulmonary rehabilitation had small negative indirect paths to HRQOL through SOB (standardized coefficient of indirect effect = −0.143, −0.049; p-values 0.000, 0.026, respectively). Age had a small indirect path to HRQOL through fatigue (standardized coefficient of indirect effect = 0.053, p-value 0.027), while supplemental oxygen use also had a small negative indirect path to HRQOL through fatigue (standardized coefficient of indirect effect = −0.091, p-value 0.000). The symptom of cough did not mediate any of the relationships between the antecedents and HRQOL.

## Discussion

### Demographic and disease characteristic impact on HRQOL

To the best of our knowledge, this is the first study that evaluated associations among demographic, disease, and symptom profiles and HRQOL in CTD-ILD. We found that while neither age nor race was associated with HRQOL, there was an association between gender and HRQOL. As hypothesized, women reported poorer HRQOL than men. This is also a finding in IPF research, in which women reported worse emotional HRQOL than men [20]. Gender differences in HRQOL were not limited to IPF. Women with rheumatoid arthritis reported poorer HRQOL than men with rheumatoid arthritis [21]. It was hypothesized that younger age was associated with worse HRQOL, but the data did not support this relationship. This may be a consequence of disease trajectory or coping mechanisms. Investigation of age and HRQOL with CTD-ILD may better be addressed through a longitudinal study during which the effects of disease trajectory can be better assessed. Another potential reason for the study finding could have to do with coping. Younger individuals with rheumatoid arthritis have better coping skills than older adults with rheumatoid arthritis [22]. Coping in chronic illness has been associated with HRQOL [23].

It was hypothesized that Black individuals with CTD-ILD would have poorer HRQOL, but this was not the case in our study. This may be a consequence of the study population. Our data showed that 78% of the individuals were White and 18% were Black, traditionally not seen in the CTD-ILD population. There is up to a threefold higher frequency of both lupus and scleroderma in Blacks in comparison to Whites [24,25]. There is also a higher prevalence of rheumatoid arthritis in Blacks compared to Whites [26]. The reasons for underrepresentation in the study population is unclear but may be a consequence of location of care. A recent study found that Whites were more than twice as likely than Blacks to receive care at an academic medical center [27].

Our study found that, while HRQOL was impaired for patients with CTD-ILD, it did not differ among types of CTD-ILD. This is a similar finding for individuals with CTD. A study by Salaffi et al. reported that HRQOL did not differ between types of CTD [28]. Our study is consistent with these findings.

We did not find a relationship between disease duration and HRQOL. This is inconsistent with findings in other types of chronic diseases. Longer disease duration was associated with worse HRQOL in individuals with rheumatoid arthritis, asthma, and COPD [29,30]. The reasons for this are not entirely clear. This may be due in part to a lack of disease progression or perhaps a successful treatment regimen. Neither of these factors were assessed in our study. The impact of adaptation may also explain the study's findings, as those with a longer disease duration may have developed the ability to adapt to their condition. In a study involving young adults with chronic illness since childhood, the ability to adapt to their illness had an impact on their HRQOL [31].

**Table 6.** Mediation Effect of Symptoms on HRQOL.

| Antecedents | Direct Effect | | Indirect Effect | | | | Overall Effect | |
|---|---|---|---|---|---|---|---|---|
| | Standardized Coefficient (SE) | *P*-value | Symptoms | Standardized Individual Coefficient (SE) | *P*-value | Standardized Overall Indirect Effect (SE) | Standardized Coefficient (SE) | *P*-value |
| Age | 0.060 (0.043) | 0.164 | Shortness of breath | −0.028 (0.023) | 0.231 | 0.027 (0.041) | 0.087 (0.057) | 0.131 |
| | | | Cough | 0.001 (0.005) | 0.763 | | | |
| | | | Fatigue | 0.053 (0.024) | 0.027 | | | |
| Gender | 0.08 (0.039) | 0.042 | Shortness of breath | 0.057 (0.022) | 0.011 | 0.091 (0.037) | 0.171 (0.053) | 0.001 |
| | | | Cough | −0.004 (0.005) | 0.369 | | | |
| | | | Fatigue | 0.039 (0.022) | 0.07 | | | |
| Race2 | 0.02 (0.041) | 0.62 | Shortness of breath | −0.044 (0.023) | 0.053 | −0.038 (0.039) | −0.018 (0.056) | 0.746 |
| | | | Cough | 0.006 (0.006) | 0.315 | | | |
| | | | Fatigue | 0 (0.022) | 0.987 | | | |
| Race3 | −0.094 (0.039) | 0.017 | Shortness of breath | 0.008 (0.022) | 0.698 | 0.019 (0.038) | −0.075 (0.054) | 0.161 |
| | | | Cough | −0.004 (0.005) | 0.441 | | | |
| | | | Fatigue | 0.014 (0.022) | 0.528 | | | |
| FVC | −0.093 (0.045) | 0.038 | Shortness of breath | 0.08 (0.027) | 0.003 | 0.104 (0.045) | 0.011 (0.063) | 0.864 |
| | | | Cough | −0.017 (0.011) | 0.146 | | | |
| | | | Fatigue | 0.04 (0.024) | 0.098 | | | |
| Supplemental Oxygen | 0.047 (0.045) | 0.303 | Shortness of breath | −0.143 (0.031) | 0 | −0.226 (0.043) | −0.18 (0.058) | 0.002 |
| | | | Cough | 0.007 (0.006) | 0.246 | | | |
| | | | Fatigue | −0.091 (0.026) | 0 | | | |
| Immunosuppressant Medication Use | 0.043 (0.037) | 0.241 | Shortness of breath | 0.032 (0.021) | 0.116 | 0.053 (0.035) | 0.096 (0.05) | 0.055 |
| | | | Cough | 0.003 (0.004) | 0.499 | | | |
| | | | Fatigue | 0.018 (0.02) | 0.377 | | | |
| Pulmonary Rehabilitation | −0.055 (0.039) | 0.16 | Shortness of breath | −0.049 (0.022) | 0.026 | −0.079 (0.036) | −0.134 (0.052) | 0.011 |
| | | | Cough | −0.003 (0.005) | 0.49 | | | |
| | | | Fatigue | −0.027 (0.021) | 0.204 | | | |
| Disease Duration 1–3 Years | 0.026 (0.047) | 0.577 | Shortness of breath | 0.007 (0.025) | 0.792 | 0.021 (0.044) | 0.109 (0.063) | 0.086 |
| | | | Cough | −0.003 (0.005) | 0.587 | | | |
| | | | Fatigue | 0.017 (0.025) | 0.495 | | | |
| Disease Duration 3–7 Years | 0.088 (0.046) | 0.058 | Shortness of breath | 0.037 (0.025) | 0.147 | 0.037 (0.025) | 0.168 (0.063) | 0.008 |
| | | | Cough | −0.008 (0.007) | 0.256 | | | |
| | | | Fatigue | 0.051 (0.026) | 0.046 | | | |
| Disease Duration >7 Years | 0.008 (0.044) | 0.852 | Shortness of breath | 0.005 (0.024) | 0.823 | 0.035 (0.042) | 0.123 (0.062) | 0.047 |
| | | | Cough | −0.008 (0.007) | 0.244 | | | |
| | | | Fatigue | 0.038 (0.024) | 0.122 | | | |
| Type of CTD-ILD: Mixed and Undifferentiated CTD | 0.096 (0.048) | 0.045 | Shortness of breath | −0.037 (0.026) | 0.159 | −0.031 (0.045) | 0.057 (0.064) | 0.371 |

*(Continued)*

**Table 6.** (Continued)

| Antecedents | Direct Effect | | Indirect Effect | | | | Overall Effect | |
|---|---|---|---|---|---|---|---|---|
| | Standardized Coefficient (SE) | P-value | Symptoms | Standardized Individual Coefficient (SE) | P-value | Standardized Overall Indirect Effect (SE) | Standardized Coefficient (SE) | P-value |
| | | | Cough | 0.004 (0.006) | 0.48 | | | |
| | | | Fatigue | 0.002 (0.026) | 0.929 | | | |
| Type of CTD-ILD: Rheumatoid Arthritis | −0.06 (0.05) | 0.236 | Shortness of breath | −0.011 (0.027) | 0.691 | 0.006 (0.047) | 0.057 (0.064) | 0.371 |
| | | | Cough | 0 (0.005) | 0.985 | | | |
| | | | Fatigue | 0.017 (0.027) | 0.539 | | | |
| Type of CTD-ILD: Sjogren's Disease | −0.02 (0.042) | 0.629 | Shortness of breath | −0.028 (0.023) | 0.226 | −0.027 (0.04) | 0.061 (0.061) | 0.32 |
| | | | Cough | 0.002 (0.005) | 0.706 | | | |
| | | | Fatigue | −0.001 (0.023) | 0.971 | | | |
| Type of CTD-ILD: Lupus | 0.031 (0.04) | 0.432 | Shortness of breath | −0.034 (0.022) | 0.13 | −0.054 (0.038) | 0.034 (0.06) | 0.575 |
| | | | Cough | 0.005 (0.005) | 0.323 | | | |
| | | | Fatigue | −0.025 (0.022) | 0.253 | | | |
| Type of CTD-ILD: Systemic Sclerosis/Scleroderma | −0.048 (0.049) | 0.324 | Shortness of breath | −0.043 (0.027) | 0.109 | −0.015 (0.046) | 0.073 (0.065) | 0.262 |
| | | | Cough | 0.003 (0.006) | 0.529 | | | |
| | | | Fatigue | 0.024 (0.026) | 0.354 | | | |
| Type of CTD-ILD: Vasculitis | −0.009 (0.038) | 0.811 | Shortness of breath | 0.005 (0.021) | 0.818 | 0.001 (0.036) | 0.088 (0.059) | 0.132 |
| | | | Cough | 0 (0.004) | 0.95 | | | |
| | | | Fatigue | −0.004 (0.021) | 0.854 | | | |

Supplemental oxygen use was associated with poorer HRQOL as hypothesized. The reason for this may be twofold, severity of disease and difficulties with oxygen equipment. The need for supplemental oxygen arises as ILD progresses causing irreversible scarring of lung tissue. Additionally, pulmonary hypertension can result from the scarring and is a common complication of CTD-ILD resulting in hypoxia [32]. Individuals with ILD that have additional comorbidities such as pulmonary hypertension have poorer HRQOL [1]. With greater severity of disease, HRQOL is impacted as shown in our study, in which lower FVC was associated with poorer HRQOL. Additionally, individuals with ILD report portable oxygen tanks to be constraining and cumbersome; using portable oxygen tanks results in not only more SOB from having to pull the tanks along as they walk, but they also walk shorter distances [33]. The cumbersome nature of oxygen equipment also leads to social isolation which further impacts HRQOL [34]. However, in a recent study, ambulatory supplemental oxygen use had a positive impact on HRQOL for individuals exertional hypoxia due to fibrotic ILD [35]. Our finding of poorer HRQOL associated with supplemental oxygen use may be a consequence of comorbid conditions present in our study population, such as pulmonary hypertension. Additionally, our study included supplemental oxygen both at rest and when ambulating. Oxygen use at rest is usually an indication of increased disease severity. The discrepancy between these two studies should not dissuade supplemental oxygen use if indicated in order to prevent sequelae of hypoxia. It was hypothesized that the use of immunosuppressant medication was associated with poorer HRQOL, based on similar findings in the renal transplant community citing side effects of immunosuppressant drugs [36,37]. An explanation for differences found in our study could be the higher number of immunosuppressant medications with more side effects that a transplant patient must take. Each medication has its own side effect profile, potentially resulting in a greater side effect burden for a

transplant patient, while those with CTD-ILD often take only one immunosuppressant medication. It was also posited that individuals with CTD-ILD taking immunosuppressant medications have poorer HRQOL possibly due to the sequelae of having to take them at a younger age, resulting in longer lifetime exposure to side effects and potential reproductive implications related to the fetal toxicity of many immunosuppressive medications. As our study was cross-sectional, duration of immunosuppressant use was not assessed. Additionally, evaluating of the effect of immunosuppressant medication use on HRQOL was not assessed by age groups.

Our study did not support the hypothesis that pulmonary rehabilitation participation is associated with improved HRQOL. This is in contrast to several systematic reviews that showed improved HRQOL and shortness of breath associated with pulmonary rehabilitation in both ILD and CTD-ILD [38]. Pulmonary rehabilitation is recommended for individuals with ILD, but patients with more severe disease are referred more frequently than those with less severe disease [39,40]. Our data demonstrate that individuals with a lower FVC, hence more severe disease, have a lower HRQOL. Therefore, our finding of poorer HRQOL associated with pulmonary rehabilitation participation may be explained by the observation that only sicker individuals with ILD are being referred to pulmonary rehabilitation, and they already have a poorer HRQOL. Additionally, as this was a cross-sectional study, it is unclear how long participants had been active in pulmonary rehabilitation. Perhaps there may have been improvement in their HRQOL after completion of a pulmonary rehabilitation program.

### Symptoms and HRQOL

As hypothesized, symptoms of SOB, cough, and fatigue were associated with HRQOL. Considering all the study variables, symptoms had the strongest association with HRQOL. These are similar findings to research in other chronic diseases such as COPD, IPF, and rheumatoid arthritis that are characterized by a high symptom burden [8,41–43]. Worsening cough severity in fibrotic ILD has been shown to be associated not only with HRQOL but also disease progression and survival [44]. In addition to cough, SOB, and fatigue, symptoms also impacting HRQOL in these other chronic diseases include depression and anxiety [8,41–43].

### Mediation effects

SOB mediated the relationships between HRQOL and gender, FVC, supplemental oxygen use, and pulmonary rehabilitation causing poorer HRQOL. The effect of SOB as a mediator between the demographic and disease characteristic on HRQOL may potentially be mitigated by minimizing the effect of SOB. There are several mechanisms that may improve SOB in ILD. For instance, pulmonary rehabilitation has demonstrated improvement with SOB in individuals with ILD [45]. Palliative care may also improve SOB. A study investigating palliative care impact on patients with refractory breathlessness due to several different chronic diseases, including ILD and COPD, found that palliative care improved the degree of SOB [46].

Fatigue was a mediator between HRQOL and supplemental oxygen such that the presence of fatigue was associated with worse HRQOL. Mitigating the effect of fatigue may affect the relationship between supplemental oxygen use and HRQOL. Pulmonary rehabilitation is recommended for individuals with ILD and has been shown to improve fatigue in both COPD and lung cancer [47,48]. Fatigue in ILD can be caused by poor sleep hygiene, which can occur in the context of obstructive sleep apnea [49]. The incidence of obstructive sleep apnea is up to 88% for individuals with ILD compared to 2%−4% for healthy adults [50]. Screening for obstructive sleep apnea in CTD-ILD patients and treating this condition as appropriate may improve fatigue.

### Conclusion

HRQOL is a critical dimension of individuals' overall wellbeing and, increasingly,a meaningful endpoint in research guiding treatment modifications and symptom relief [51]. Healthcare intervention effectiveness has historically been based

on biomedical outcomes (survival, disease severity) rather than HRQOL [51]. A need for changes in ILD trial design to include endpoints such as how patients feel, function, and survive has been identified as a priority [52]. Most research involving CTD-ILD is currently focused on interventions affecting FVC, as opposed to HRQOL. In fact, there is no composite PROM that takes into account both rheumatological and pulmonary disease impact on HRQOL. It is important to consider an intervention's impact not just on a patient's physical perspective but also their emotional and social well-being as they live with their disease. Our data indicate that individuals with CTD-ILD have poor HRQOL. Factors associated with diminished HRQOL in this population include lower FVC, oxygen use, pulmonary rehabilitation participation, female gender, and symptoms of SOB, cough, and fatigue. The observed correlation between symptoms and HRQOL provides evidentiary support for symptom-targeted interventions. Importantly, these findings also suggest that palliative care-traditionally underutilized in this population-may offer a viable approach to improving HRQOL. Future research should focus on developing and evaluating palliative care interventions tailored to CTD-ILD, with HRQOL as a central outcome measure. Identifying high risk subgroups and addressing modifiable symptoms may ultimately lead to meaningful improvements in HRQOL for this patient population.

## Author contributions

**Conceptualization:** Lanier L. O'Hare, Anand S. Iyer, Pariya L. Wheeler, Liang Shan, Marie Bakitas.

**Data curation:** Lanier L. O'Hare, Anand S. Iyer, Kathleen O. Lindell.

**Formal analysis:** Lanier L. O'Hare, Kathleen O. Lindell, Liang Shan, Marie Bakitas.

**Investigation:** Lanier L. O'Hare, Anand S. Iyer, Kathleen O. Lindell, Pariya L. Wheeler.

**Methodology:** Lanier L. O'Hare, Anand S. Iyer, Kathleen O. Lindell, Liang Shan, Marie Bakitas.

**Project administration:** Lanier L. O'Hare, Pariya L. Wheeler.

**Supervision:** Lanier L. O'Hare, Anand S. Iyer, Pariya L. Wheeler, Liang Shan, Marie Bakitas.

**Validation:** Lanier L. O'Hare, Kathleen O. Lindell, Pariya L. Wheeler, Liang Shan.

**Writing – original draft:** Lanier L. O'Hare.

**Writing – review & editing:** Lanier L. O'Hare, Anand S. Iyer, Kathleen O. Lindell, Pariya L. Wheeler, Liang Shan, Tracy Luckhardt, Marie Bakitas.

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
