## [Decision Letter · Decision Letter 0]

PONE-D-25-04818Factors Affecting Quality of Life in Connective Tissue Disease-Related Interstitial Lung DiseasePLOS ONE

Dear Dr. Ohare,

Thank you for submitting your manuscript to PLOS ONE. After careful consideration, we feel that it has merit but does not fully meet PLOS ONE’s publication criteria as it currently stands. Therefore, we invite you to submit a revised version of the manuscript that addresses the points raised during the review process.

Specifically, our reviewers found some interests in this study, but pointed out a number of issues that require improvement. I ask the authors to fully respond to all comments made by reviewers in the revised manuscript. 

We look forward to receiving your revised manuscript.

Kind regards,

Masataka Kuwana, MD, PhD

Academic Editor

PLOS ONE

Journal Requirements:

2. Please note that your Data Availability Statement is currently missing the DOI/accession number of each dataset OR a direct link to access each database. If your manuscript is accepted for publication, you will be asked to provide these details on a very short timeline. We therefore suggest that you provide this information now, though we will not hold up the peer review process if you are unable.

Reviewers' comments:

Reviewer's Responses to Questions

**Comments to the Author**

1. Is the manuscript technically sound, and do the data support the conclusions?

Reviewer #1: Yes

Reviewer #2: Yes

Reviewer #3: Yes

2. Has the statistical analysis been performed appropriately and rigorously? 

Reviewer #1: No

Reviewer #2: Yes

Reviewer #3: Yes

3. Have the authors made all data underlying the findings in their manuscript fully available?

Reviewer #1: No

Reviewer #2: Yes

Reviewer #3: Yes

4. Is the manuscript presented in an intelligible fashion and written in standard English?

Reviewer #1: Yes

Reviewer #2: Yes

Reviewer #3: Yes

5. Review Comments to the Author

Reviewer #1: This is an interesting study that investigated the quality of life and its associated factors in patients with CTD-ILD. This is an important study but there are several things to be clarified.

1. The rationale that the authors selected SF-6D to assess quality of life in patients with CTD-ILD should be clarified. Although SF-6D is a good and representative tool to assess QOL, patients with CTD have lots of symptoms other than ILD.

2. Only 66.1% of the participants were treated with immunosuppressive medications. I wonder if those patients would be really representative of CTD-ILD. And before that, what was the definition of immunosuppressive medications? If that include glucocorticoids, synthetic immunosuppressive drugs, and also biological agents, the details should be described.

3. Demographics and disease characteristics of the participants should be tabulated in one table at first.

4. “The type of ILD” should be “the type of CTD” in Table 4.

5. The authors should explain and discuss more clearly what clinical impact this study would make.

Reviewer #2: Manuscript ID PONE-D-25-04818

"Factors Affecting Quality of Life in Connective Tissue Disease-Related Interstitial Lung Disease"

General comments

This cross-sectional study utilizing a secondary data analysis from the Pulmonary Fibrosis Foundation Patient Registry (PFFPR) investigated the association between patient demographic and disease characteristics, symptoms, and QOL in CTD-ILD. As the authors stated, the impact of CTD-ILD on QOL has yet to be sufficiently described due to the paucity of information about QOL in this population. Therefore, this study is important as the patient-centered outcomes research. Especially, examination for the mediation effects is interesting and important.

1. Quality of life (QOL) and health-related quality of life (HRQoL)

Quality of life (QOL) is a broad term defined as an individual’s wellbeing in relation to their goals and expectations, in the context of the cultural and value systems they live in (WHO). Health-related quality of life (HRQoL) is a subset of QOL that describes an individual’s perception of the impact of health, disease, and treatment on their QOL.

The authors used Medical Outcome Study Short Form Six-Dimension Health Survey (SF-6D) as one of the patient reported outcome measures (PROMs) in this study. SF-6D, as well as the Medical Outcome Study 36-Item Short Form Health Survey (SF-36), is a generic measure for HRQoL. Most papers the authors cited in this manuscript are related to HRQoL rather than QOL. Considering that HRQoL refers to the impact of physical and mental health on an individual’s QOL, the authors should be caution in use of the term ‘QOL’. I recommend to use the term ‘HRQoL’ instead of ‘QOL’ in many parts in this manuscript.

2. Significance of supplemental oxygen use and pulmonary rehabilitation

According to the results of this study the authors mentioned that supplemental oxygen use was associated with poorer QOL and there was a statistically significant difference in QOL with those participating in pulmonary rehabilitation reporting poorer QOL.

I am worried about the negative impact of supplemental oxygen use and pulmonary rehabilitation in clinical practice for patients with CTD-ILD. I recommend to add some comments on these points in the DISCUSSION section as I pointed out in the Minor comments.

3. Study population regarding race

The authors stated that 78% of the individuals were White and 18% were Black, traditionally not seen in the CTD-ILD population. As the authors mentioned in the DISCUSSION section, in the United States, Black individuals often have less access to specialized healthcare, which may have influenced the study’s inclusion process and potentially led to selection bias. How about the percentage of White and Black in the original PFFPR population? Are there any differences in the percentage of White and Black between the original PFFPR population and the CTD-ILD population in this study?

Minor comments

INTRODUCTION

P3

・The mortality rate for ILD has increased over 50% in the past 10 years and is the 40th most common cause of death.2

The authors should add the information about countries and territories in this study.2 This study, the Global Burden of Disease 2015 Study (GBD 2015), provides a comprehensive assessment of all-cause and cause-specific mortality for 249 causes in 195 countries and territories from 1980 to 2015.

・Connective tissue disease (CTD) is responsible for approximately one third of all cases.4

The paper you cited as #4 (Cochrane Database Syst Rev. 2018;1(1): CD010908) is the Cochrane Review for cyclophosphamide therapy for CTD-ILD. The correct paper you should cite here is ‘Mittoo S, Gelber AC, Christopher-Stine L, Horton MR, Lechtzin N, Danoff SK. Ascertainment of collagen vascular disease in patients presenting with interstitial lung disease. Respiratory Medicine 2009;103:1152-8.’

The paper #4 describes as follows; ‘Approximately 30% of individuals with ILD have associated CTD, which presents subsequent to the development of ILD in about 15% of these individuals (Mittoo S, Gelber AC, Christopher-Stine L, Horton MR, Lechtzin N, Danoff SK. Ascertainment of collagen vascular disease in patients presenting with interstitial lung disease. Respiratory Medicine 2009;103:1152-8.).

RESULTS

Descriptive Analyses

P7

・Most participants did not require supplemental oxygen (206 [61.9%]), were not taking immunosuppressant medications (220 [66.1%]),・・・

Could the authors describe the type of immunosuppressant medications; glucocorticoids, other disease-modifying anti-rheumatic drugs (DMARDs) or other immunomodulatory agents, in the study population? Although the authors mentioned that QOL was not associated with immunosuppressive medication use, in patients with rheumatoid arthritis the impact of glucocorticoids on HRQoL has been investigated from the perspective of the patient (Cheah JTL, et al. The patient's perspective of the adverse effects of glucocorticoid use: A systematic review of quantitative and qualitative studies. From an OMERACT working group. Semin Arthritis Rheum. 2020;50(5):996-1005.). Our readers may want to know the type of immunosuppressant medications.

DISCUSSION

P16

・A recent study found that Whites were more than twice as likely than Blacks to receive care at an academic medical center.27

These findings could be changed by different countries or territories. The authors should add the information about territories studied in this study, that is, Boston and New York City.27

・Supplemental oxygen use was associated with poorer QOL as hypothesized.

I recommend to add some comments on the effects of supplemental oxygen on HRQoL in ILD patients using the following paper showing that ambulatory oxygen seems to be associated with improved HRQoL in patients with ILD.

Visca D, et al. Effect of ambulatory oxygen on quality of life for patients with fibrotic lung disease (AmbOx): a prospective, open-label, mixed-method, crossover randomised controlled trial. Lancet Respir Med. 2018;6(10): 759-770.

P17

・It was hypothesized that the use of immunosuppressant medication was associated with poorer QOL, based on similar findings in the renal transplant community citing side effects of immunosuppressant drugs.35,36

The authors described the different situations for immunosuppressant medication use between CTD-ILD patients and renal transplant patients.

As mentioned above, in patients with rheumatoid arthritis the impact of glucocorticoids on HRQoL has been investigated (Cheah JTL, et al. The patient's perspective of the adverse effects of glucocorticoid use: A systematic review of quantitative and qualitative studies. From an OMERACT working group. Semin Arthritis Rheum. 2020;50(5):996-1005.). In addition, in patients with sarcoidosis, glucocorticoids can reportedly negatively affect HRQoL (Judson MA, et al. The effect of corticosteroids on quality of life in a sarcoidosis clinic: the results of a propensity analysis. Respir Med. 2015;109(4):526-31. Cox CE, et al. Health-related quality of life of persons with sarcoidosis. Chest 2004;125(3):997-1004.). Do you have any additional comments on the effects of immunosuppressant medication, especially for glucocorticoids, for HRQoL in CTD-ILD patients? CTD-ILD and sarcoidosis may have similar situations for immunosuppressant medication use.

P18

・Perhaps there may have been improvement in their QOL after completion of a pulmonary rehabilitation program.

I recommend to add some comments about the effects of pulmonary rehabilitation for ILD on HRQoL using a paper showing that pulmonary rehabilitation probably improves HRQoL (Dowman L, Hill CJ, May A, Holland AE. Pulmonary rehabilitation for interstitial lung disease. Cochrane Database Syst Rev. 2021; 2(2): CD006322.). In addition, recently published systematic review for pulmonary rehabilitation in CTD-ILD shows that pulmonary rehabilitation benefits moderate levels of evidence for quality of life in CTD-ILD patients (Seleoglu I, Demirel A. Pulmonary rehabilitation in connective tissue disease-associated interstitial lung disease: A systematic review. Sarcoidosis Vasc Diffuse Lung Dis. 2024; 41(4): e2024061).

・Mediation Effects

The authors reported that the symptom of cough did not mediate any of the relationships between the antecedents and QOL in the RESULTS section.

I recommend to add some comments on these results because this may signify the importance of cough as one of the PROMs. Worse cough severity is reportedly independently associated with worse HRQoL in fibrotic ILD (Khor YH, et al; CARE-PF Investigators. Epidemiology and prognostic significance of cough in fibrotic interstitial lung disease. Am J Respir Crit Care Med. 2024; 210(8): 1035-1044.)

Reviewer #3: This is a very valuable paper that considers the quality of life of patients with CTD-ILD. There are no particular problems with the theme or the main content, and I think it is an important study in today's world where patient-centred medicine is talked about.

<minor comments="">

In this study, FVC and supplemental oxygen are being examined, but I wonder if DLco is not being examined?

Also, can you mention whether or not Shortness of breath in Symptom A occurs during exertion, such as during a six-minute walk test? If there is CTD, there is a possibility that there are latent vascular lesions, for example in scleroderma, and SOB during exertion may have a stronger impact on QOL.

P2L16: paticipants → participants

P2L25: pallpiative → palliative</minor>

6. PLOS authors have the option to publish the peer review history of their article (what does this mean? ). If published, this will include your full peer review and any attached files.

**Do you want your identity to be public for this peer review?** For information about this choice, including consent withdrawal, please see our Privacy Policy .

Reviewer #1: No

Reviewer #2: **Yes: ** Hiromi Tomioka

Reviewer #3: No

---

## [Author Response · Author response to Decision Letter 1]

16 May 2025

Manuscript PONE-D-25-04818

Response to Reviewers

Dear Dr. Kuwana,

Thank you for providing the opportunity to submit a revised draft of our manuscript “Factors Affecting Quality of Life in Connective Tissue Disease-Related Interstitial Lung Disease” for publication in PLOS ONE. We are appreciative of the time and expertise that the reviewers dedicated to providing feedback on our manuscript. We are particularly grateful for the insightful comments provided for the improvement of our paper. The reviewer suggestions have been incorporated into the manuscript. Please see below for a point by point response to the reviewers’ comments. All page numbers refer to the revised manuscript with tracked changes.

Reviewer’s Comments to the Authors:

Reviewer #1

This is an interesting study that investigated the quality of life and its associated factors in patients with CTD-ILD. This is an important study but there are several things to be clarified.

Author response: Thank you and I agree!

1. The rationale that the authors selected SF-6D to assess quality of life in patients with CTD-ILD should be clarified. Although SF-6D is a good and representative tool to assess QOL, patients with CTD have lots of symptoms other than ILD.

Author response: While the SF-6D is an effective measure for QOL assessment, it does not directly address the additional rheumatological symptoms associated with CTD-ILD. There are patient reported outcome measures (PROM) that evaluate the impact of CTD on QOL. One such validated measure is the EQ-5D that considers the impact of rheumatoid arthritis on mobility, self-care, usual activities, pain, depression, and anxiety (Rabin R, de Charro F. EQ-5D: a measure of health status from the EuroQol Group. Ann Med. 2001;33(5):337-343). As this study was a secondary analysis of data collected for the Pulmonary Fibrosis Foundation Patient Registry (PFFPR), it is limited by the PROM utilized by the that organization. The only PROM addressing QOL in the PFFPR was the SF-6D. Please see the revised manuscript addressing this as a study limitation (p. 19).

2. Only 66.1% of the participants were treated with immunosuppressive medications. I wonder if those patients would be really representative of CTD-ILD. And before that, what was the definition of immunosuppressive medications? If that include glucocorticoids, synthetic immunosuppressive drugs, and also biological agents, the details should be described.

Author response: The immunomodulatory agents in this study include abatacept, adalimumab, azathioprine, belimumab, cyclophosphamide, cyclosporin A, etanercept, golimumab, hydroxychloroquine, infliximab, leflunomide, methotrexate, mycophenolate, rituximab, sulfasalazine, tacrolimus, and tocilizumab. This information is included in the revised manuscript. In terms of the study finding of 66.1% of participants being treated with immunosuppressive agents being representative of CTD-ILD, this data is difficult to ascertain for CTD-ILD in general. There is a study that investigated telomere length and immunosuppression in non-idiopathic pulmonary fibrosis interstitial lung disease found that 49% of participants with CTD-ILD took immunosuppressive medications (Zhang D, Adegunsoye A, Oldham JM, et al. Telomere length and immunosuppression in non-idiopathic pulmonary fibrosis interstitial lung disease. Eur Respir J. 2023;62(5):2300441). However, some types of CTD are associated with worse ILD and this study did not differentiate amongst types of CTD. There is more discreet data with specific types of CTD-ILD. A study evaluating immunosuppression use in ILD due to scleroderma found 71% of individuals with scleroderma-related ILD took immunosuppression (Adler S, Huscher D, Siegert E, et al. Systemic sclerosis associated interstitial lung disease - individualized immunosuppressive therapy and course of lung function: results of the EUSTAR group. Arthritis Res Ther. 2018;20(1):17). In rheumatoid arthritis-related ILD, one study found immunosuppressant use in 65% of patients (Li, L., Liu, R., Zhang, Y., Zhou, J., Li, Y., Xu, Y., ... & Zheng, Y. (2020). A retrospective study on the predictive implications of clinical characteristics and therapeutic management in patients with rheumatoid arthritis-associated interstitial lung disease. Clinical rheumatology, 39, 1457-1470.) Our study finding of 66.1% treatment rate for CTD-ILD to be in line with specific CTD conditions.

3. Demographics and disease characteristics of the participants should be tabulated in one table at first.

Author response: Demographic and disease characteristics of the participants are tabulated on page 8, Table 3. Table 4 is now a comparison of HRQOL by categorical variables, Table 5 is now the correlation coefficients, and Table 6 is the mediation effect of symptoms on HRQOL.

4. “The type of ILD” should be “the type of CTD” in Table 4.

Author response: Noted and corrected in the manuscript. The change is reflected in revised Table 3 and Table 4.

5. The authors should explain and discuss more clearly what clinical impact this study would make.

Author response: Additional clinical impact and discussion added to pages 20-21.

Reviewer #2

This cross-sectional study utilizing a secondary data analysis from the Pulmonary Fibrosis Foundation Patient Registry (PFFPR) investigated the association between patient demographic and disease characteristics, symptoms, and QOL in CTD-ILD. As the authors stated, the impact of CTD-ILD on QOL has yet to be sufficiently described due to the paucity of information about QOL in this population. Therefore, this study is important as the patient-centered outcomes research. Especially, examination for the mediation effects is interesting and important.

Author response: Thank you!

Reviewer 2 General Comments

1. Quality of life (QOL) and health-related quality of life (HRQoL)

Quality of life (QOL) is a broad term defined as an individual’s wellbeing in relation to their goals and expectations, in the context of the cultural and value systems they live in (WHO). Health-related quality of life (HRQoL) is a subset of QOL that describes an individual’s perception of the impact of health, disease, and treatment on their QOL.

The authors used Medical Outcome Study Short Form Six-Dimension Health Survey (SF-6D) as one of the patient reported outcome measures (PROMs) in this study. SF-6D, as well as the Medical Outcome Study 36-Item Short Form Health Survey (SF-36), is a generic measure for HRQoL. Most papers the authors cited in this manuscript are related to HRQoL rather than QOL. Considering that HRQoL refers to the impact of physical and mental health on an individual’s QOL, the authors should be caution in use of the term ‘QOL’. I recommend to use the term ‘HRQoL’ instead of ‘QOL’ in many parts in this manuscript.

Author response: I agree with this correction, as the study did, indeed, evaluate HRQoL. The manuscript has been corrected to reflect this change.

2. Significance of supplemental oxygen use and pulmonary rehabilitation

According to the results of this study the authors mentioned that supplemental oxygen use was associated with poorer QOL and there was a statistically significant difference in QOL with those participating in pulmonary rehabilitation reporting poorer QOL.

I am worried about the negative impact of supplemental oxygen use and pulmonary rehabilitation in clinical practice for patients with CTD-ILD. I recommend to add some comments on these points in the DISCUSSION section as I pointed out in the Minor comments.

Author response: Supplemental oxygen use and pulmonary rehabilitation were associated with poorer QOL in this study and I agree that the implications of these findings could cause concern in clinical practice, perhaps resulting in hesitation to pursue either treatment. A more robust discussion of these findings has been added to the discussion section, specifically addressing additional confounding factors that may account for these findings (p. 17, 18).

3. Study population regarding race

The authors stated that 78% of the individuals were White and 18% were Black, traditionally not seen in the CTD-ILD population. As the authors mentioned in the DISCUSSION section, in the United States, Black individuals often have less access to specialized healthcare, which may have influenced the study’s inclusion process and potentially led to selection bias. How about the percentage of White and Black in the original PFFPR population? Are there any differences in the percentage of White and Black between the original PFFPR population and the CTD-ILD population in this study?

Author response: In our study, the race of the CTD-ILD study population (78% White and 18% Black), is somewhat similar to the PFFPR population in which 84% are White, but only 5% Black. Of note, 55% of Black individuals in the entire PFFPR cohort are included in the CTD-ILD cohort.

Reviewer 2 Minor Comments

INTRODUCTION

P3

・The mortality rate for ILD has increased over 50% in the past 10 years and is the 40th most common cause of death.2

The authors should add the information about countries and territories in this study.2 This study, the Global Burden of Disease 2015 Study (GBD 2015), provides a comprehensive assessment of all-cause and cause-specific mortality for 249 causes in 195 countries and territories from 1980 to 2015.

Author response: The manuscript has been revised to include specific information about geographic regions (deaths in 2015: Central Asia 1705, Central Europe 1806, Latin America 4856, and the United States 18722).

・Connective tissue disease (CTD) is responsible for approximately one third of all cases.4

The paper you cited as #4 (Cochrane Database Syst Rev. 2018;1(1): CD010908) is the Cochrane Review for cyclophosphamide therapy for CTD-ILD. The correct paper you should cite here is ‘Mittoo S, Gelber AC, Christopher-Stine L, Horton MR, Lechtzin N, Danoff SK. Ascertainment of collagen vascular disease in patients presenting with interstitial lung disease. Respiratory Medicine 2009;103:1152-8.’

The paper #4 describes as follows; ‘Approximately 30% of individuals with ILD have associated CTD, which presents subsequent to the development of ILD in about 15% of these individuals (Mittoo S, Gelber AC, Christopher-Stine L, Horton MR, Lechtzin N, Danoff SK. Ascertainment of collagen vascular disease in patients presenting with interstitial lung disease. Respiratory Medicine 2009;103:1152-8.).

Author response: The proper citation for the percentage of ILD cases involving CTD has been added.

RESULTS

Descriptive Analyses

P7

・Most participants did not require supplemental oxygen (206 [61.9%]), were not taking immunosuppressant medications (220 [66.1%]),・・・

Could the authors describe the type of immunosuppressant medications; glucocorticoids, other disease-modifying anti-rheumatic drugs (DMARDs) or other immunomodulatory agents, in the study population? Although the authors mentioned that QOL was not associated with immunosuppressive medication use, in patients with rheumatoid arthritis the impact of glucocorticoids on HRQoL has been investigated from the perspective of the patient (Cheah JTL, et al. The patient's perspective of the adverse effects of glucocorticoid use: A systematic review of quantitative and qualitative studies. From an OMERACT working group. Semin Arthritis Rheum. 2020;50(5):996-1005.). Our readers may want to know the type of immunosuppressant medications.

Author response: The immunomodulatory agents in this study include abatacept, adalimumab, azathioprine, belimumab, cyclophosphamide, cyclosporin A, etanercept, golimumab, hydroxychloroquine, infliximab, leflunomide, methotrexate, mycophenolate, rituximab, sulfasalazine, tacrolimus, and tocilizumab. This information has been added to the manuscript (p. 4, 5).

DISCUSSION

P16

・A recent study found that Whites were more than twice as likely than Blacks to receive care at an academic medical center.27

These findings could be changed by different countries or territories. The authors should add the information about territories studied in this study, that is, Boston and New York City.27

Author response: In terms of the finding of a recent study that Whites were more than twice as likely than Blacks to receive care at an academic medical center; I agree that this could be affected by different countries or territories. Unfortunately, there was no granular geographic data such as cities collected by the PFFPR. The only geographic data collected was regional in the United States: West, Midwest, Northeast, and South. Each region is comprised of urban, suburban, and rural areas but this is not subcategorized in the PFFPR.

Supplemental oxygen use was associated with poorer QOL as hypothesized.

I recommend to add some comments on the effects of supplemental oxygen on HRQoL in ILD patients using the following paper showing that ambulatory oxygen seems to be associated with improved HRQoL in patients with ILD.

Visca D, et al. Effect of ambulatory oxygen on quality of life for patients with fibrotic lung disease (AmbOx): a prospective, open-label, mixed-method, crossover randomised controlled trial. Lancet Respir Med. 2018;6(10): 759-770.

Author response: I agree that it is important to point out that supplemental oxygen has been shown to improve HRQoL in patients with ILD and have added this in the discussion section (p. 18).

P17

・It was hypothesized that the use of immunosuppressant medication was associated with poorer QOL, based on similar findings in the renal transplant community citing side effects of immunosuppressant drugs.35,36

The authors described the different situations for immunosuppressant medication use between CTD-ILD patients and renal transplant patients.

As mentioned above, in patients with rheumatoid arthritis the impact of glucocorticoids on HRQoL has been investigated (Cheah JTL, et al. The patient's perspective of the adverse effects of glucocorticoid use: A systematic review of quantitative and qualitative studies. From an OMERACT working group. Semin Arthritis Rheum. 2020;50(5):996-1005.). In addition, in patients with sarcoidosis, glucocorticoids can reportedly negatively affect HRQoL (Judson MA, et al. The effect of corticosteroids on quality of life in a sarcoidosis clinic: the results of a propensity analysis. Respir Med. 2015;109(4):526-31. Cox CE, et al. Health-related quality of life of persons with sarcoidosis. Chest 2004;125(3):997-1004.). Do you have any additional comments on the effects of immunosuppressant medication, especially for glucocorticoids, for HRQoL in CTD-ILD patients? CTD-ILD and sarcoidosis may have similar situations for immunosuppressant medication use.

Author response: Both the positive effect of glucocorticoids on disease activity and negative impact on HRQoL in CTD and sarcoidosis is well documented. I would expect to see similar findings in the CTD-ILD population, however, glucocorticoid use was not included as immunosuppression in the PFFPR database.

P18

Perhaps there may have been improvement in their QOL after completion of a pulmonary rehabilitation program.

I recommend to add some comments about the effects of pulmonary rehabilitation for ILD on HRQoL using a paper showing that pulmonary rehabilitation probably improves HRQoL (Dowman L, Hill CJ, May A, Holland AE. Pulmonary rehabilitation for interstitial lung disease. Cochrane Database Syst Rev. 2021; 2(2): CD006322.). In addition, recently published systematic review for pulmonary rehabilitation in CTD-ILD shows that pulmonary rehabilitation benefits moderate levels of evidence for quality of life in CTD-ILD patients (Seleoglu I, Demirel A. Pulmonary rehabilitation in connective tissue disease-associated interstitial lung disease: A systematic review. Sarcoidosis Vasc Diffuse Lung Dis. 2024; 41(4): e2024061).

Author response: I agree that discussion of our finding that pulmonary rehabilitation is associated with poorer HRQoL should occur in the context of prior research showing it probably improves HRQoL. This has been added to the manuscript (p. 19). 

---

## [Decision Letter · Decision Letter 1]

Factors Affecting Quality of Life in Connective Tissue Disease-Related Interstitial Lung Disease

PONE-D-25-04818R1

Dear Dr. Ohare,

We’re pleased to inform you that your manuscript has been judged scientifically suitable for publication and will be formally accepted for publication once it meets all outstanding technical requirements.

Kind regards,

Masataka Kuwana, MD, PhD

Academic Editor

PLOS ONE

Additional Editor Comments (optional):

Reviewers' comments:

Reviewer's Responses to Questions

**Comments to the Author**

1. If the authors have adequately addressed your comments raised in a previous round of review and you feel that this manuscript is now acceptable for publication, you may indicate that here to bypass the “Comments to the Author” section, enter your conflict of interest statement in the “Confidential to Editor” section, and submit your "Accept" recommendation.

Reviewer #1: All comments have been addressed

Reviewer #2: All comments have been addressed

2. Is the manuscript technically sound, and do the data support the conclusions?

Reviewer #1: Yes

Reviewer #2: Yes

3. Has the statistical analysis been performed appropriately and rigorously? 

Reviewer #1: Yes

Reviewer #2: Yes

4. Have the authors made all data underlying the findings in their manuscript fully available?

Reviewer #1: Yes

Reviewer #2: Yes

5. Is the manuscript presented in an intelligible fashion and written in standard English?

Reviewer #1: Yes

Reviewer #2: Yes

6. Review Comments to the Author

Reviewer #1: (No Response)

Reviewer #2: General comments

The authors replied appropriately and revised the manuscript.

Minor comments

KEY WORDS:

quality of life→health-related quality of life

Table 1. Variable

Quality of life→Health-related quality of life

7. PLOS authors have the option to publish the peer review history of their article (what does this mean? ). If published, this will include your full peer review and any attached files.

**Do you want your identity to be public for this peer review?** For information about this choice, including consent withdrawal, please see our Privacy Policy .

Reviewer #1: No

Reviewer #2: **Yes: ** Hiromi Tomioka

---

## [Editor Report · Acceptance letter]

PONE-D-25-04818R1

PLOS ONE

Dear Dr. Ohare,

I'm pleased to inform you that your manuscript has been deemed suitable for publication in PLOS ONE. Congratulations! Your manuscript is now being handed over to our production team.

Kind regards,

on behalf of

Prof. Masataka Kuwana

Academic Editor

PLOS ONE